# Effect of Plasma Nitriding Pretreatment on the Mechanical Properties of AlCrSiN-Coated Tool Steels

**DOI:** 10.3390/ma12050795

**Published:** 2019-03-07

**Authors:** Yin-Yu Chang, Siddhant Amrutwar

**Affiliations:** Department of Mechanical and Computer-Aided Engineering, National Formosa University, Yunlin 632, Taiwan; siddhant.amrutwar@gmail.com

**Keywords:** duplex treatment, plasma nitriding, hard coating, tool steel

## Abstract

Surface modification of steel has been reported to improve hardness and other mechanical properties, such as increase in resistance, for reducing plastic deformation, fatigue, and wear. Duplex surface treatment, such as a combination of plasma nitriding and physical vapor deposition, achieves superior mechanical properties and resistance to wear. In this study, the plasma nitriding process was conducted prior to the deposition of hard coatings on the SKH9 substrate. This process was done by a proper mixture of nitrogen/hydrogen gas at suitable duty cycle, pressure, and voltage with proper temperature. Later on, the deposition of gradient AlCrSiN coatings synthesized by a cathodic-arc deposition process was performed. During the deposition of AlCrSiN, CrN, AlCrN/CrN, and AlCrSiN/AlCrN were deposited as gradient interlayers to improve adhesion between the coatings and nitrided steels. A repetitive impact test (200k–400k times) was performed at room temperature and at high temperature (~500 °C) to assess impact resistance. The results showed that the tribological impact resistance for the synthesized AlCrSiN increased because of a progressive hardness support. The combination of plasma nitriding and AlCrSiN hard coatings is capable of increasing the life of molding dies and metal forging dies in mass production.

## 1. Introduction

For warm stamping of stainless steels and light alloys, the molds are operated at high temperature—thermal and load cycling—and often lose effectiveness early because of oxidation and wear. The plasma nitriding process (PN) is a well-known technique, which can be utilized to improve the surface properties of ferrous alloys [1,2]. Similar to PN, physical vapor deposition (PVD) techniques make coatings useful for tool steels with good tribological performance due to high hardness, anti-abrasion, and high-temperature oxidation resistance. However, their mechanical performance is restricted when they are deposited on a soft substrate due to plastic deformation under high loads without adequate support. Duplex treatments including PN and PVD are verified to be successful in creating wear and corrosion resistance with good mechanical support. The adhesion strength between the coating and the tool steel can be controlled by nitriding techniques and gradient coating designs to tailor the nitrided layer and by inserting a surface treatment method, such as polishing, before the deposition of hard coatings [3,4,5,6]. 

Currently, PVD hard coatings such as CrN, TiN, CrAlN, and TiAlN are used to improve the properties of tool steels and extend the lifetime of mechanical components and tools [7,8,9,10,11]. The introduction of silicon into CrAlN could decrease the grain size, change the coating structure, composition, and its mechanical properties [12,13,14]. In this CrAlSiN, silicon is apt to segregate into amorphous SiN_x_ along the grain boundaries. Many nanocomposite and nanolayered coatings have been studied due to their excellent mechanical and tribological properties. The coating structure and chemical content of the film can influence the mechanical and tribological properties of multicomponent coatings in the form of gradient and multilayered layers. Previous studies have shown that multilayered and graded coatings, such as AlCrSiN/TiVN and TiVN/TiSiN, are very effective in reducing wear in molding, forming, and machining applications [15,16,17]. High-speed steels and hot-working steels are the universal stamping, cutting, and machining tool steels because of their good strength at high temperature and are usually used for metal forming, cutting tools, and light alloy injection. However, these tool steels have poor resistance to tribological wear with a high coefficient of friction. The demand has been increased using surface modification and coating techniques to improve the mechanical and wear performances of the tool steel. Thermal-and-chemical surface modification such as PN is applied to these steels for prolonging the lifetime of tools [18]. This study tried to develop plasma nitriding before the AlCrSiN hard coating deposition to improve the mechanical properties of tool steel substrates and the adhesion strength. An impact fatigue test was conducted at room temperature and at high temperature. The purpose was to study the microstructure, mechanical properties, and periodic impact resistance of the high-speed steels with duplex treatment combing PVD AlCrSiN coatings and PN. 

## 2. Materials and Methods

The heat-treated SKH9 tool steels (25 mm in diameter and 4 mm in thickness) having hardness of 60 HRC (Rockwell Hardness scale C) were used as substrates. A nominal chemical composition of the SKH9 steel in wt.% was as follows: 0.8–0.9 wt.% C, 3.8–4.6 wt.% Cr, 5.5–7 wt.% W, 0.45 wt.% Mn, 1.5–2.2 wt.% V, 4.5–5.5 wt.% Mo, 4.3–5.2 wt.% Co, and Fe as balance. The samples were metalographically prepared using a grinding machine (SiC emery paper up to 1500-grit size, Jet measurement Co., Taichung, Taiwan) and a polishing machine (suspension with Al_2_O_3_ particle size 0.05 μm, Jet measurement Co., Taichung, Taiwan) to have a surface roughness of Ra = 0.2 ± 0.05 μm. The tool steel samples were cleaned using ultrasonic agitation in an alcoholic solution for 15 min. After that, the samples were dried by hot air. The PN treatment was conducted using an industrial ion and plasma nitriding system, as shown in Figure 1a. The reactor was equipped with a rotary pump to obtain low vacuum before the PN process. Ar was introduced and the samples were sputtered prior to plasma nitriding to remove the native oxide layer and keep clean. Afterwards, nitrogen/hydrogen mixture was introduced to form a nitride layer on samples. The detailed processing parameters are shown in Table 1 [19,20]. After PN treatment, the tool steel samples were cooled down in the furnace to the room temperature under vacuum conditions. The nitriding process usually raises the surface roughness. In this study, the as-nitrided tool steels had a surface roughness of Ra = ~0.5 μm, and the subsequent polishing after PN decreased the surface roughness effectively. The nitrided samples were polished prior to PVD to have Ra = 0.2 ± 0.05 μm. The PVD technique followed by cathodic-arc evaporation used different types of targets: AlCrSi alloy, AlCr alloy, and pure chromium (Cr), as shown in Figure 1b. Table 2 shows the deposition parameters of AlCrSiN. The deposition process was initially Ar (99.999% pure) ion bombardment followed by the deposition of a Cr adhesion layer. The introduction of nitrogen into the chamber then led to the formation of CrN on top of the Cr layer. CrN was used as the bottom interlayer, and then CrN/AlCrSiN transition layers were deposited. The contents of Al and Si increased to form an AlCrSiN top layer. The temperature of the PN treatment and PVD was measured near the specimen holder using a thermocouple whose thermal junction was insulated from the holder with a quartz cup.

A field emission scanning electron microscope (FESEM, JSM IT-100, JEOL Ltd., Tokyo, Japan), which was equipped with an energy-dispersive X-ray spectroscopy (EDS) system, was used to investigate the coating morphology and microstructure. The texture and phase identification of the AlCrSiN with and without PN-treated samples were examined by X-ray diffraction MXP III MAC Science (Bruker’s) with Cu radiation. The XRD system was operated at 40 kV, using a low glancing incidence angle of 4° to identify the coating structure. The X-ray beam incidence angle was fixed while the XRD detector was moved in the 2θ between 20° and 90° to obtain the diffraction pattern of the samples. 

The adhesion strength of the PN and PN + AlCrSiN-coated samples was evaluated by Rockwell C indentation (HR-200, Mitutoyo Co., Japan) according to the ISO 26443 standard. The hardness of the films and PN-treated substrates was measured by the Vickers tester (MMT-X, Matsuzawa Co., Akita, Japan) with a load of 25 g and measured by an average value from seven measured points. In this study, a Calotest equipment was used to determine the thickness of the hard coatings. The coated samples were placed under an optical microscope to show a circular crater shape and then measure the coating thickness [21]. 

To study the impact fatigue resistance of the deposited AlCrSiN coatings with and without PN-pretreated samples, a cyclic impact test was conducted using a periodic loading device at room temperature and at a high temperature of 500 °C [22]. The distance between the sample and indenter and the impact frequency were set at 1 mm and 20 Hz, respectively. Continuous impacts of a reciprocating WC–Co ball indenter were applied onto the deposited AlCrSiN with and without PN. The test was performed without lubrication under a load of 9.8 N. 

## 3. Results and Discussion

### 3.1. Microstructure Characterization

The nitrided layer is usually composed of the compound layer and the diffusion layer. The compound zone mainly consists of porous ε-nitrides, whereas the diffusion layer is nitric martensite [23,24]. Duplex treatments combine PN with the subsequent deposition of hard coatings to create good tribological properties. In this study, samples of plasma nitrided SKH9 steel, AlCrSiN-coated sample, and PN + AlCrSiN (with PN pretreatment) coated SKH9 steel were tested for research. Figure 2 shows typical X-ray diffraction (XRD) patterns of plasma nitrided (PN), AlCrSiN-coated, and PN + AlCrSiN-coated samples. After plasma nitriding, it can be seen that the XRD patterns of the nitrided compound layer consists of diffraction peaks of CrN, Fe_x_N, where x = 2–3, and Fe_4_N, as shown in Figure 2a. The substrate α′-martensite peak at 82.4° was detected in the PN-treated layer, and it suggested that the nitrided layer was thin and less than the X-ray penetration depth, and resulted in observing the substrate α′-martensite phase. The main nitride precipitates that were identified were Fe_2–3_N and Fe_4_N. These precipitates play the main role in increasing the surface hardness and improving the wear resistance, thereby, increasing the time of fatigue fracture [25]. For the AlCrSiN, when N_2_ was introduced during the coating process, the Al, Cr, and Si plasma generated by arc evaporators induced the excitation of N_2_ to form AlCrSiN on the substrate. Figure 2b shows the XRD pattern of the AlCrSiN without plasma nitriding treatment. Regardless of the low Si content (less than 10 at.%), the sequence of crystalline phases was observed, and no corresponding peaks of crystalline silicon nitride were found. The crystallography structure of single-phase face-centered cubic (FCC) AlCrN solid solution was detected. The four main peaks correspond to reflections (111), (200), (220), and (311) consisting of nanocrystalline FCC-AlCrN [26,27,28]. Figure 2c shows the diffraction pattern of PN + AlCrSiN consisting of AlCrN and Fe_4_N phases. The Fe_4_N compound layer was obtained during the nitriding process. The compound layer led to higher hardness and improved the adhesion quality of coatings. Similar to the previous AlCrSiN without plasma nitriding, the AlCr(Si)N phases exhibited cubic B1-NaCl structure. The lattice structure of the deposited coatings showed solid solution in these coatings. The lattice parameter of the AlCrSiN was 0.424 nm, which was between the values of cubic AlN (JCPDF file No.: #251495) and CrN (JCPDF file No.: #110065). The microstructure was obtained by the formation of an FCC phase, obtained by the substitution of Al and Si in the cubic CrN phase. The Si-containing AlCrSiN coatings had better oxidation and tribological resistances than CrN and CrAlN and possessed good mechanical properties [14,29,30]. 

In order to study the microstructure of coating construction, Figure 3 shows the fractured cross-sectional SEM image of the AlCrSiN coating on an Si (100) wafer. Deposited at a high bias voltage of −120 V, the top AlCrSiN layer showed fine dense structures. CrN and AlCrN/CrN bottom interlayers showed obvious columnar structure. After AlCrSiN was co-deposited with AlCrN, the formation of the columnar structure was reduced. The deposited AlCrSiN with a transition layer of AlCrSiN/AlCrN possessed a much denser structure because of the presence of Al and Si to form compact nanocrystalline structures. Similar results by He et al. [31] also showed that the single-phase cubic CrAlN coatings exhibited a columnar structure, while dense and featureless characteristic morphologies for AlCrSiN coatings were observed. The total thickness of the graded AlCrSiN coating was 2.24 μm, which was similar to the result of Calotest (~2 μm), and consisted of the top AlCrSiN layer, transition layer of AlCrSiN/AlCrN, and columnar AlCrN/CrN interlayers. From the EDS measurement, the chemical composition of the deposited AlCrSiN top layer coating was 26.7 at.% Al, 17.6 at.% Cr, 4.2 at.% Si, and 51.5 at.% N. It showed atomic stoichiometries of Al_0.55_Cr_0.37_Si_0.08_N for the deposited AlCrSiN. Distinct interfaces were observed at the boundaries between the transition layer (AlCrSiN/AlCrN) and the columnar AlCrN/CrN interlayers. In this AlCrSiN, the addition of Si impeded grain growth and led to grain re-nucleation, and. Therefore. resulted in a compact and dense structure. The retardation of columnar growth by the incorporation of Si in the CrAlN coating was proved, and previous studies demonstrated that the overall oxidation resistance of the AlCrSiN coatings after Si doping was significantly improved at high temperature [32,33,34].

### 3.2. Mechanical Properties

#### 3.2.1. Hardness Test

High surface hardness of the tool steels can increase capacities of load support and high abrasion wear resistance. In this study, the measurement was performed using a Vickers microhardness tester. The applied load on the specimen was 25 gf. Figure 4 shows the hardness of SKH9 high-speed steel substrate, PN-treated, AlCrSiN-coated, and PN + AlCrSiN-coated samples under different processes. The hardness for PN samples was 1352 HV_25g_, which was higher than that of SKH9 high-speed steel. From the results, the PN + AlCrSiN possessed hardness of about 3256 HV_25g_, which was the highest hardness among all the samples, and the AlCrSiN-coated samples (without PN pretreatment) possessed lower hardness, i.e., 2285 HV_25g_, than PN + AlCrSiN. The hardness of the PN + AlCrSiN-coated samples was 2.4 times higher than that of the PN-treated SKH9. In this case, it revealed that the AlCrSiN-coated sample with PN pretreatment could achieve the highest hardness among the samples. The duplex treatment combining PN and hard coating exhibited higher hardness as compared to non-duplex-treated samples. The nitrided layer was helpful for the progressive hardness distribution, inducing the gradual transition of mechanical properties improving the wear resistance [35,36]. 

#### 3.2.2. Adhesion Test

The adhesion strength of all the PN, AlCrSiN-, and PN + AlCrSiN-coated samples was investigated using the Rockwell indentation test. This destructive test, may exhibit two distinctive properties of the coated compound [37,38]. We used Rockwell indentation with a load of 150 kg based on the ISO 26443 standard to evaluate the adhesion strength of the coatings. According to the criteria, class 0 reveals acceptable adhesion. Class 1 shows no adhesive delamination; adhesion is acceptable. In the cases of class 2 and class 3, adhesion is unacceptable [39]. Adhesive delamination is defined as a removal of the coating, whereby the underlying substrate can be clearly seen, or a removal of one or more sublayers in a multilayered or graded coating, whereby the substrate or an underlying sublayer can be clearly distinguished. Figure 5 shows optical images to reveal the crack behavior of the AlCrSiN- and PN + AlCrSiN-coated samples. The optical image with high magnification shows the crack condition on the edge of the indented crater. The AlCrSiN coating (without PN) showed that, initially, circumferential cone cracks developed beneath the indenter, followed by radial and annular cracks as well as some coating spallation, which appeared at the edge of the indenter crater. The poor adhesion of class 2 is recognized on contrast. The coating was too stiff to buckle, which created the formation of compressive stress cracks at the wear interface of the coating and the specimen. These compressive stress-induced cracks propagated, and the formation of spallation in coatings occurred. This kind of adhesion failure was usually observed in brittle ceramic coatings with weak interfacial adhesion to the specimen. The plastic deformation easily took place due to its low hardness, resulting in the cracks formed in the indentation edge. If the interfacial adhesion of coatings was large in area, lesser wedge spallation occurred on the substrate coatings. 

A significant change was observed in the PN + AlCrSiN-coated samples. In case of this study, PN + AlCrSiN coatings showed no cohesive failure and no coating delamination, which meant it had no cracks and good adhesion properties of class 0 due to a hard nitrided layer before the deposition of AlCrSiN coating on the substrate. The hardness was increased by the nitrided layer, which provided enough support for the hard coatings even under heavy loads. This fact was also confirmed by He and Deng et al. [35,40]. Surface hardening by plasma nitriding improved the heavier load bearing capacity for AlCrSiN-coated specimens. The nitrided layer provided good adhesion property and was useful in sustaining a high load. It may be helpful for the improvement of the adhesion on PN + AlCrSiN SKH9 substrate in this study. 

#### 3.2.3. Impact Fatigue Test

The impact tester was designed to simulate fatigue wear in tribological coating systems subjected to a dynamic stress on the coated samples [41]. In this study, we explored the anti-fatigue properties and impact resistance of the coated samples using a room temperature fatigue test and a high-temperature fatigue test (500 °C). The number of impacts was 200,000 (200k), 300,000 (300k), and 400,000 (400k). It was tested using a dynamic impact fatigue tester for the AlCrSiN- (without PN pretreatment) and the PN + AlCrSiN-coated samples. During the test, a tungsten carbide ball with a diameter of 2 mm was used as a punch. The frequency was controlled at 20 Hz and a load of 9.8 N was applied. The film was subjected to periodic reciprocating impact tests, and then the film was observed for damage, sticky and stacked conditions, and bare substrate. Referring to the previous study by Batista et al. [42], the damage of the film after the impact test was studied. During the process of the impact fatigue test, deformation of the surface takes place step by step depending upon the impacts and load. The failure zones can be divided into three types: Central zone with cohesive failure, intermediate zone with adhesion and cohesive failure, and lastly peripheral failure, which usually takes place at the boundary of the indentation area. Figure 6 shows the SEM image and EDS element mapping of the AlCrSiN film at room temperature after 200,000 (200k), 300,000 (300k), and 400,000 (400k) impacts. EDS mapping revealed that a composition signal with various compositions such as aluminum (Al), chromium (Cr), silicon (Si), tungsten (W), iron (Fe), and nitrogen (N) signals was obtained. After continuous cycles of the loading and unloading process, the surface morphology was observed through element mapping. The AlCrSiN-coated samples showed smooth morphology, where no significant transfer of the WC–Co ball material was observed after 200k impacts. Chipping outside of the impacted crater was usually the main failure for ceramic hard coatings in the impact fatigue test [22]. However, in this study there were no significant changes for the wear morphology with increasing impacts from 200k to 400k cycles. But in case of impact cavity size under a continuous process of loading–unloading at room temperature and the normal load of 9.8 N after 300k impacts, formation of the cohesive and adhesive failures of the AlCrSiN film in the intermediate and central zones of the cavity was observed. No iron (Fe) signal was observed on the surface. This indicated the substrate was not exposed at this stage. The results showed that the AlCrSiN-coated samples can sustain the load of around 400k impacts at room temperature. This phenomenon indicated that, after 400k continuous impacts at room temperature, the AlCrSiN-coated sample had possible formation of cohesive failure, which might later effect the hardness and cause failures in the mechanical performance of dies. 

Figure 7 reveals the results of the PN + AlCrSiN-coated sample at the same impacts of 200k, 300k, and 400k at room temperature showing the surface is not yet exposed at this stage. No obvious substrate iron (Fe) signal can be found. The benefit of the duplex treatment in terms of improving the load bearing capacity of hard coatings originated from its higher critical load than those from non-duplex-treated parts. The change in crater volume with an increase in the number of impact cycles for the PN + AlCrSiN duplex-treated sample had no significant change observed. As the impact cycles increased from 200k to 400k there was an absence of rapid increases in wear as compared to the non-duplex AlCrSiN sample. This might be the reason for the increase in load of coatings and minimizing substrate deformation, which reduced the bending and stretching of the coatings. A similar result of impact wear resistance of duplex PN-treated/PVD-coated Ti–6Al–4V alloy was revealed by Cassar et al. [43]. 

The high-temperature fatigue impact test at 500 °C was performed on the AlCrSiN-coated high-speed SKH9 steel. The SEM image with EDS element mapping image is shown in Figure 8. The elemental mappings of aluminum (Al), chromium (Cr), silicon (Si), tungsten (W), oxygen (O), iron (Fe), and nitrogen (N) signals are shown. It showed that the surface was exposed of iron signal from the substrate of the AlCrSiN-coated SKH9 steel. Therefore the substrate was exposed to 200k impacts at the high temperature of 500 °C. On the other hand, a high-temperature impact of PN + AlCrSiN after 200k impacts was conducted. As shown in Figure 9, no iron (Fe) signal was present on the impacted surface. The duplex-treated sample (PN + AlCrSiN) had the presence of an Fe–N compound and a diffusion layer beneath the hard AlCrSiN coating. These two layers supported the top AlCrSiN layer from heavy mechanical loads at high temperature, which also improved the hardness and adhesion strength to the tool steel. The surface modification of SKH9 steels by a combination of plasma nitriding and coatings resulted in excellent mechanical load bearing capacity and strong adhesion at high temperature. Compared to the non-duplex-treated AlCrSiN sample that only resists 200k impacts at high temperature, duplex-treated AlCrSiN coatings had better performance of periodic impact fatigue resistance at high temperature. 

Previous studies showed that AlCrSiN possessed good thermal stability and oxidation resistance because of the formation of the epitaxial growth structure, which inhibited atomic diffusion. The nanocomposite coatings synthesized by amorphous silicon nitride could provide a stable phase and limited grain growth in nanocrystal AlCrN at high temperature [44,45,46]. However, in this study, the non-duplex-treated AlCrSiN sample only resisted 200k impacts at 500 °C. The high-temperature fatigue impact test for the PN + AlCrSiN further was carried out by continuous impacts from 300k to 400k. No obvious iron signal was detected after 300k impacts. As shown in Figure 10, after 400k impacts, it was found that the substrate failed and the iron substrate was exposed. It is concluded that the dense PN + AlCrSiN coating with good thermal stability can resist around 400k impacts at a high temperature of 500 °C. The results showed the PN + AlCrSiN-coated tool steels exhibit an excellent result in both room and high-temperature fatigue impacts. 

## 4. Conclusions

In this study, the AlCrSiN hard coatings were deposited on the plasma nitrided SKH9 steel using a duplex treatment combing plasma nitriding and cathodic-arc evaporation PVD technology. The deposited AlCrSiN had bottom interlayers of CrN and AlCrN and a transition layer (AlCrSiN/AlCrN). From SEM and XRD evaluation, it was observed that the AlCrSiN coating with gradient layer structure was a dense structure of nanocrystalline B1-NaCl, and it showed that the coating had obvious columnar bottom layers of AlCrN and CrN structure. AlCrSiN-coated samples (without PN pretreatment) possessed hardness of 2285 HV_25g_. The PN + AlCrSiN had the highest hardness of 3256 HV_25g_ among all the samples. This significant change occurred due to the support of the modified layer formed as a result of the nitriding process and the reduction of plastic deformation under the applied load. 

Impact fatigue tests for AlCrSiN- and PN + AlCrSiN-coated samples at room temperature and at high temperature (500 °C) were conducted. Both the AlCrSiN and duplex-treated samples possessed good impact fatigue performance at room temperature. The PN + AlCrSiN duplex-treated sample could resist 400k impacts at a high temperature of 500 °C, where only small area of iron signal from the substrate was exposed and no cohesive failure was observed. The PN + AlCrSiN duplex-treated tool steel had high hardness, and it possessed good impact fatigue performance at room temperature and at high temperature. 

## Figures and Tables

**Figure 1 materials-12-00795-f001:**
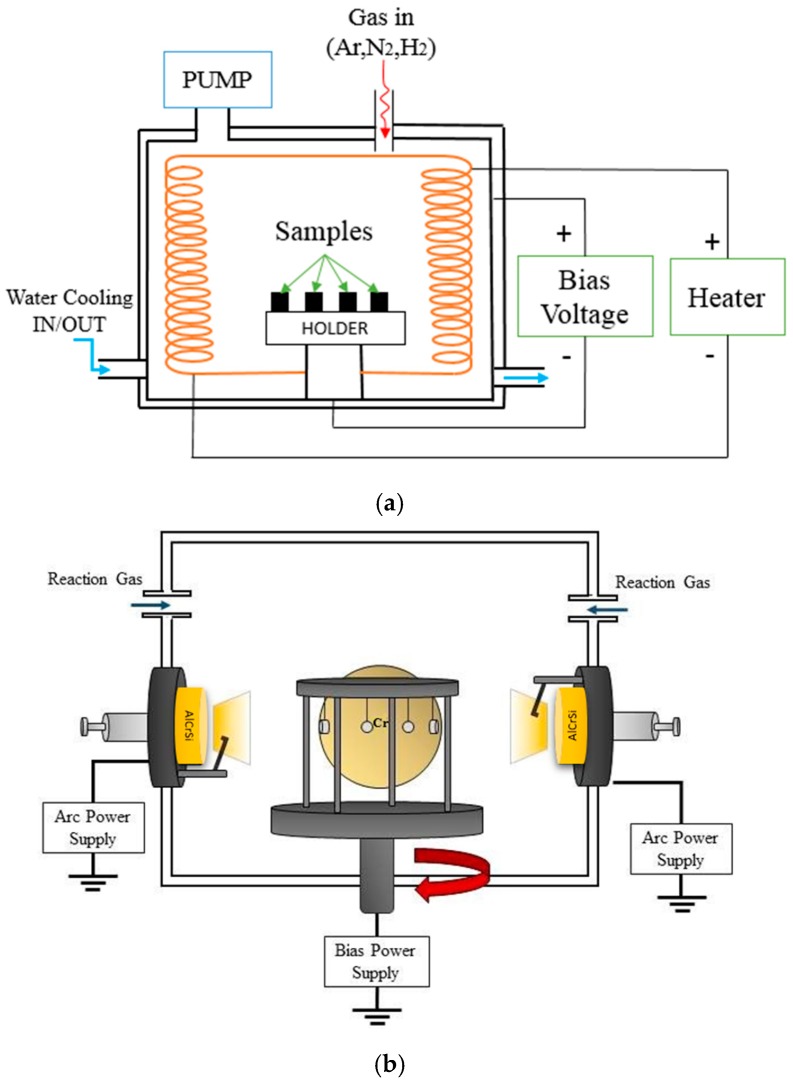
(**a**) Schematic diagram of plasma nitriding setup and (**b**) schematic diagram of physical vapor deposition (PVD) cathodic-arc evaporation system.

**Figure 2 materials-12-00795-f002:**
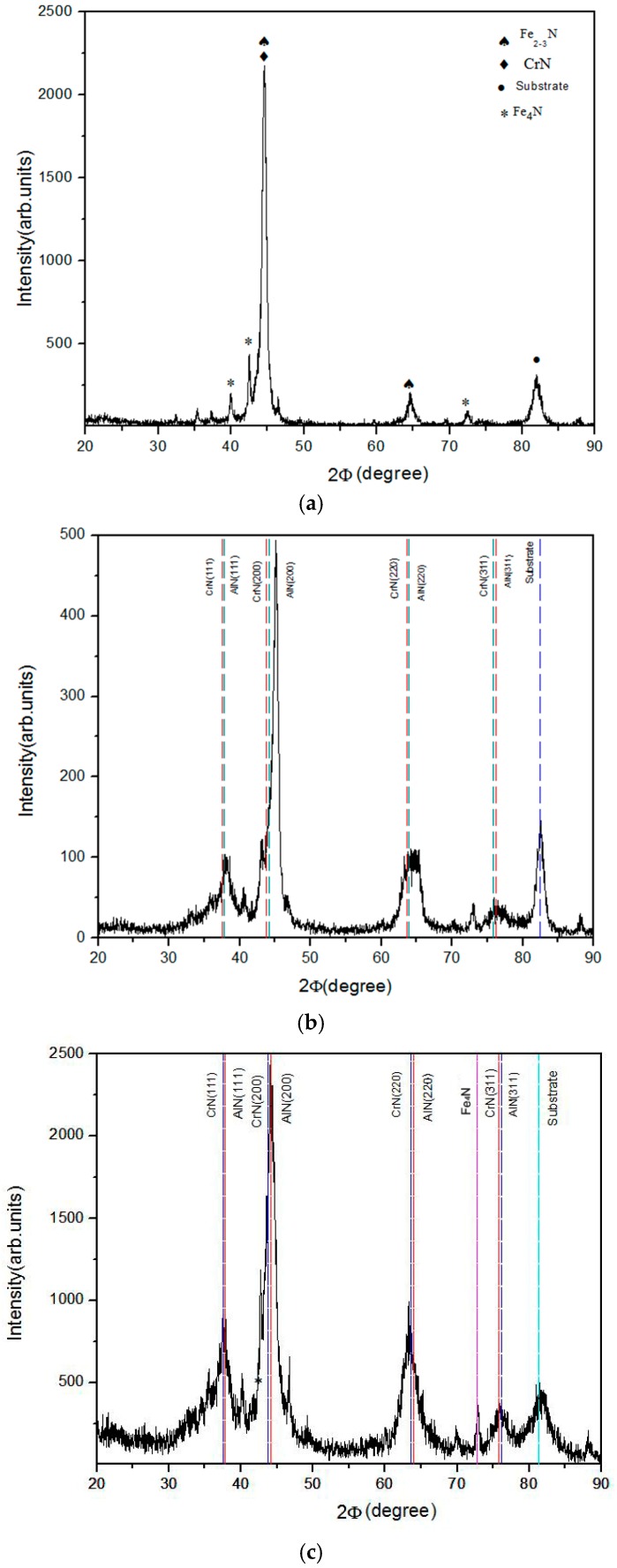
XRD pattern of (**a**) plasma nitriding (PN), (**b**) AlCrSiN, and (**c**) PN + AlCrSiN samples.

**Figure 3 materials-12-00795-f003:**
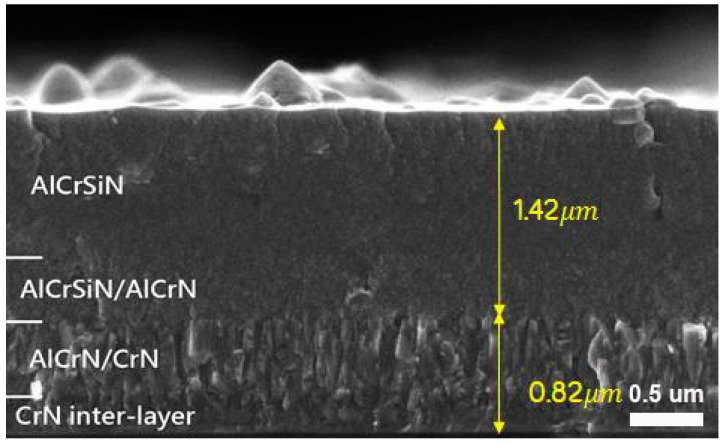
Cross-sectional SEM image of AlCrSiN coating.

**Figure 4 materials-12-00795-f004:**
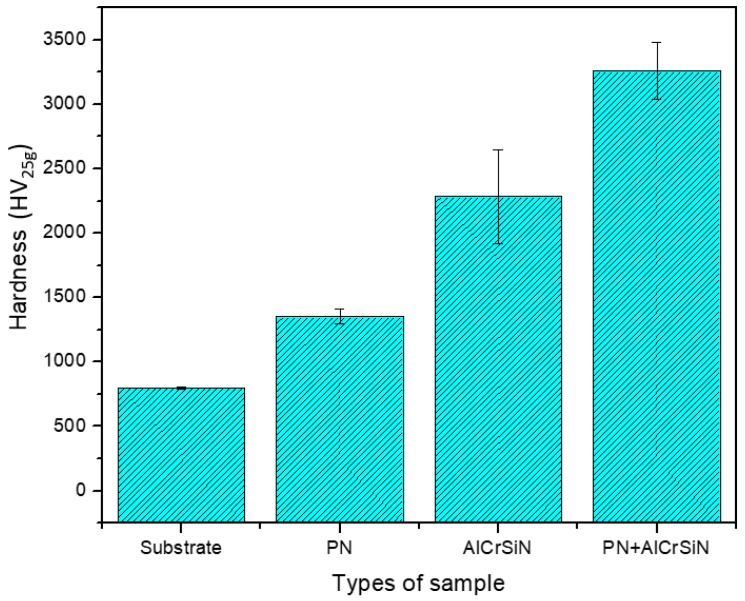
Vickers microhardnesses of the SKH9 high-speed steel substrate, PN-treated, AlCrSiN-coated, and PN + AlCrSiN-coated samples.

**Figure 5 materials-12-00795-f005:**
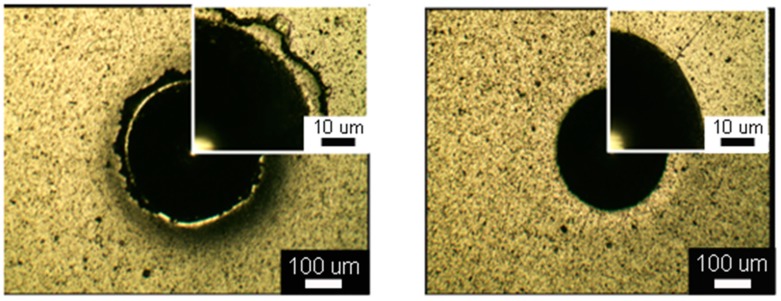
Optical images showing crack behavior evaluated by Rockwell indentation test applied on the AlCrSiN (**Left**) and PN + AlCrSiN (**Right**) coated samples.

**Figure 6 materials-12-00795-f006:**
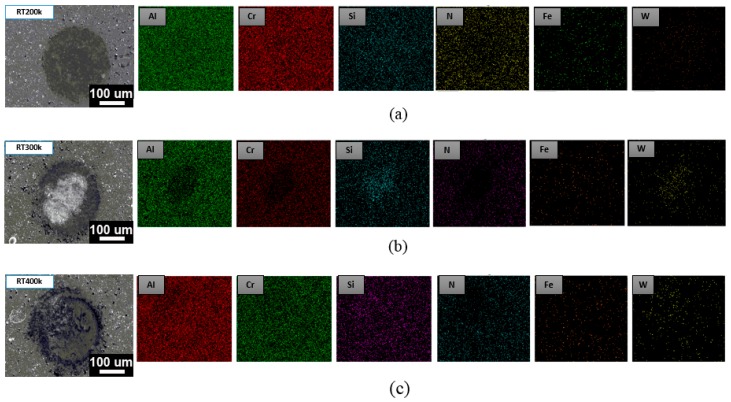
SEM and energy-dispersive X-ray spectroscopy (EDS) element mapping of AlCrSiN-coated samples at room temperature after (**a**) 200k, (**b**) 300k, and (**c**) 400k impacts.

**Figure 7 materials-12-00795-f007:**
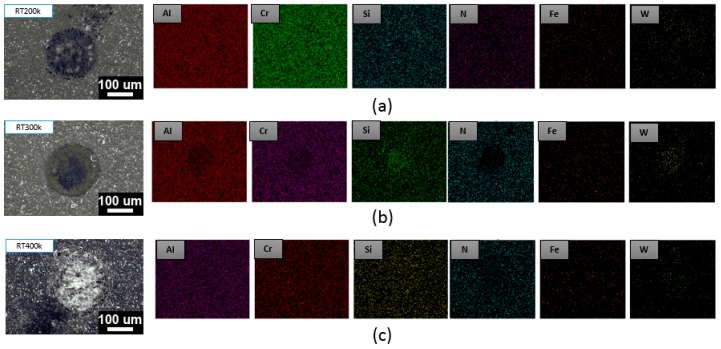
SEM and EDS element mapping of PN + AlCrSiN at room temperature after (**a**) 200k, (**b**) 300k, and (**c**) 400k impacts.

**Figure 8 materials-12-00795-f008:**
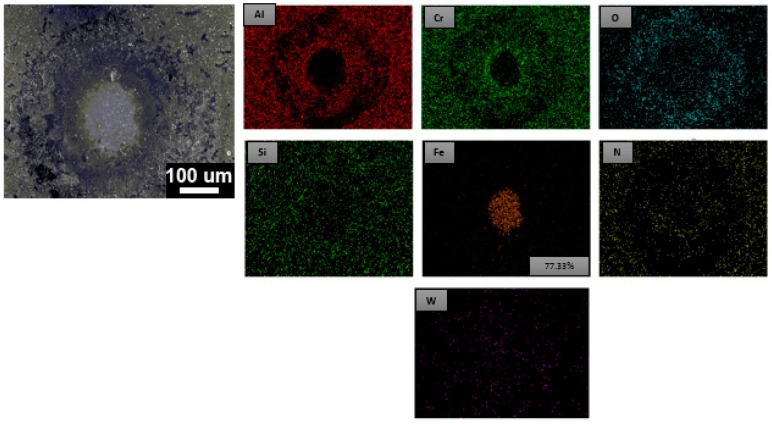
SEM and EDS element mapping of AlCrSiN at high temperature (500 °C) after 200k impacts.

**Figure 9 materials-12-00795-f009:**
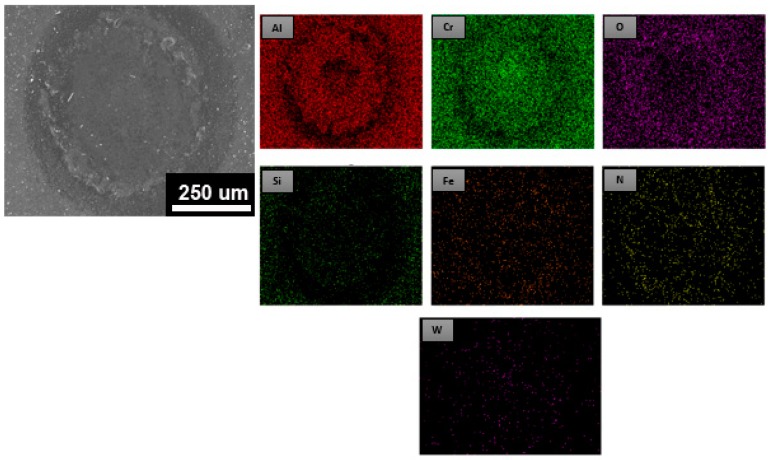
SEM and EDS element mapping of PN + AlCrSiN at high temperature (500 °C) after 200k impacts.

**Figure 10 materials-12-00795-f010:**
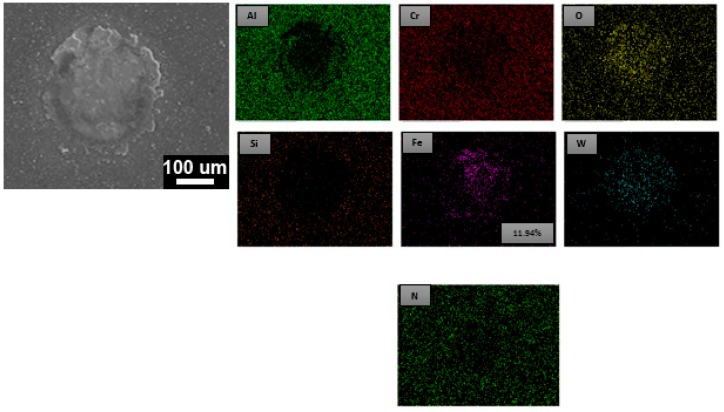
SEM and EDS element mapping of PN + AlCrSiN at high temperature (500 °C) after 400k impacts.

**Table 1 materials-12-00795-t001:** Parameters for plasma nitriding.

Parameters	
Base pressure	<1 Pa
Ar ion cleaning	Bias voltage = −700 V, Pressure = 2 Pa, Time = 30 min
Nitriding	T = 450 °C, Time = 7 h, Bias voltage = −500 V, N_2_/H_2_ = 80/20 sccm
Duty Cycle	82%
Pressure	250 Pa

**Table 2 materials-12-00795-t002:** Parameters for AlCrSiN thin-film deposition.

Parameters	
Target	Cr, Al_0.7_Cr_0.3_, Al_0.6_Cr_0.3_Si_0.1_
Base pressure	<0.004 Pa
Ar ion cleaning	Bias voltage = −700 V, Pressure = 2 Pa, Time = 20 min
Reactive gas	Ar/N_2_
Working pressure	3 Pa
Deposition temperature	250 °C
Target current	70 A
Bias voltage	−120 V
Rotation speed	2 rpm
Deposition time	90 min

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
