# Peer review of "Effect of Plasma Nitriding Pretreatment on the Mechanical Properties of AlCrSiN-Coated Tool Steels"

_materials, 2019, doi:10.3390/ma12050795_

Reviewer 1 Report

1. Information about processing regimes is repeated in section 2 and in Table 1. It is proposed to transfer all data to the Tables 1 and 2.

2. It is necessary to add in the Section 2 samples dimensions, steel composition as well as the polishing conditions and the value of the surface roughness.

3. Was the temperature of the samples measured? You need to specify the point of temperature measurement in the chamber.

4. Pressure values are given in various units (mbar and Pa).

5. Formation of martensite is unclear (page 5). Sample hardening was not performed; the martensitic phase is not shown on the diffraction pattern.

6. It is necessary to eliminate repetitions in the text. For example, “The main nitride precipitates were identified as Fe2-3N and Fe4N” (page 5).

7. The letter (b) can be removed in the caption of Fig. 5.

8. Scale marks are not visible in Fig. 6–10. 

Author Response

1. Information about processing regimes is repeated in section 2 and in Table 1. It is proposed to transfer all data to the Tables 1 and 2.

Response: Thanks for Reviewer’s suggestion. We have revised the processing description of PN treatment. Table 1 and Table are revised. All data are put into the Tables 1 and Table 2 without repetitions.

2. It is necessary to add in the Section 2 samples dimensions, steel composition as well as the polishing conditions and the value of the surface roughness.

Response: Thanks for Reviewer’s suggestion. We have revised to show samples dimensions, steel composition as well as the polishing conditions and the value of the surface roughness in section 2 Materials and Methods. The heat-treated SKH9 tool steels (25 mm in diameter and 4 mm in thickness) having hardness of 60 HRC were used as substrates. A nominal chemical composition of the SKH9 steel in wt.% was as follows: 0.8~0.9 wt.%C, 3.8~4.6 wt.% Cr, 5.5~7 wt.% W, 0.45 wt.% Mn, 1.5~2.2 wt.% V, 4.5~5.5 wt.% Mo, 4,3~5.2 wt.% Co, and Fe as balance. The samples were metalographically prepared using a grinding machine (SiC emery paper up to 1500-grit size) and a polishing machine (suspension with Al2O3 particle size 0.05 μm) to have surface roughness of Ra= 0.2± 0.05 μm.

3. Was the temperature of the samples measured? You need to specify the point of temperature measurement in the chamber.

Response: Thanks for Reviewer’s suggestion. We have revised the manuscript. The temperature of the PN treatment and PVD was measured near the specimen holder using a thermocouple whose thermal junction was insulated from the holder with a quartz cup.

4. Pressure values are given in various units (mbar and Pa).

Response: Thanks for Reviewer’s suggestion. We have revised the manuscript. The unit of mbar is replaced by Pa.

5. Formation of martensite is unclear (page 5). Sample hardening was not performed; the martensitic phase is not shown on the diffraction pattern.

Response: Thanks for Reviewer’s suggestion. We have revised the manuscript. Actually, the substrate α’- martensite peak at 82.4o was detected in nitrided layer, suggesting that the nitrided layer was thin and less than the penetration depth of X-ray, resulting in observing the substrate α’- martensite phase.

6. It is necessary to eliminate repetitions in the text. For example, “The main nitride precipitates were identified as Fe2-3N and Fe4N” (page 5).

Response: Thanks for Reviewer’s suggestion. We have revised the manuscript. The sentence “The main nitride precipitates were identified are Fe2-3N and Fe4N.” was removed to eliminate repetitions in the text.

7. The letter (b) can be removed in the caption of Fig. 5.

Response: Thanks for Reviewer’s suggestion. The letter (b) was removed in the caption of Fig. 5.

8. Scale marks are not visible in Fig. 6–10.

Response: Thanks for Reviewer’s suggestion. We have revised the scale marks in Fig. 6–10.

Reviewer 2 Report

In this paper, the authors describe the effect of plasma nitriding of SKH9 steel and deposition of a nanocrystalline layer of AlCrSiN using interlayers of CrN and AlCrN forming a steel with improved hardness and enhanced fatigue performance. I have only a few minor comments:

1. In the methods section, what do the authors mean by hot wind? Is it hot air or hot nitrogen?

2. Argon within a sentence should be written as argon (in the methods section).

3. What were the surface roughness of the nitrided samples before and after polishing?

4. In table 2, please correct the third row containing Ar/N2.

5. FCC is used before it is defined on page 5.

6. What type and crystal orientation of Si was used for the deposition? In general, the surface and orientation of the Si wafer affects the orientation of the deposition of the layers.

7. Please provide the units of hardness in Figure 4.

8. Units of pressures are mixed up between Pa and mbar, please use a consistent style.

9. The conclusion is too long, it should be shortened to provide only a summary of the work and potentials that can be explored in one paragraph only.

Author Response

1. In the methods section, what do the authors mean by hot wind? Is it hot air or hot nitrogen?

Response: Thanks for Reviewer’s suggestion. We have revised the manuscript. The specimens were cleaned in an alcoholic solution with ultrasonic agitation for 15mins at room temp. After that, specimens were dried by hot air.

2. Argon within a sentence should be written as argon (in the methods section).

Response: Thanks for Reviewer’s suggestion. We have revised the manuscript. Argon within a sentence was written as argon (Ar).

3. What were the surface roughness of the nitrided samples before and after polishing?

Response: Thanks for Reviewer’s suggestion. The nitriding process usually raises the surface roughness. The as-nitrided specimen exhibited the maximum average roughness Ra~ 0.5 μm, and the intermediate polishing after nitriding could lower the Ra effectively. The nitrided samples were polished prior to PVD to have Ra= 0.2± 0.05 μm.

4. In table 2, please correct the third row containing Ar/N2.

Response: Thanks for Reviewer’s suggestion. It was corrected as Ar/N2 in Table 2.

5. FCC is used before it is defined on page 5.

Response: Thanks for Reviewer’s suggestion. We have revised the manuscript. FCC is used before it is defined.

6. What type and crystal orientation of Si was used for the deposition? In general, the surface and orientation of the Si wafer affects the orientation of the deposition of the layers.

Response: Thanks for Reviewer’s suggestion. We have revised the manuscript. Si (100) wafer was used in this study.

7. Please provide the units of hardness in Figure 4.

Response: Thanks for Reviewer’s suggestion. The unit of hardness was provided in Figure 4.

8. Units of pressures are mixed up between Pa and mbar, please use a consistent style.

Response: Thanks for Reviewer’s suggestion. We have revised the manuscript. The unit of mbar is replaced by Pa.

9. The conclusion is too long, it should be shortened to provide only a summary of the work and potentials that can be explored in one paragraph only.

Response: Thanks for Reviewer’s suggestion. We have revised the manuscript. The conclusion was shortened to provide only a summary of the work.

Reviewer 3 Report

The article is interesting and well thought out. However, in attached PDF file I have given some proposal of corrections before printing.

Author Response

The article is interesting and well thought out. However, in attached PDF file I have given some proposal of corrections before printing.

Response: Thanks for Reviewer’s suggestion. We have revised the manuscript according to the comments in the PDF, and the changes and modifications of the manuscript are highlighted in RED colour. For example, the unit of mbar is replaced by Pa; Fe2-3N is replaced by FexN, where x= 2-3.